# A 10-Year Review on Advancements in Identifying and Treating Intellectual Disability Caused by Genetic Variations

**DOI:** 10.3390/genes15091118

**Published:** 2024-08-24

**Authors:** Kexin Hou, Xinyan Zheng

**Affiliations:** School of Exercise and Health, Shanghai University of Sport, 200 Hengren Road, Yangpu, Shanghai 200438, China; 2321517013@sus.edu.cn

**Keywords:** intellectual disability, genetics, chromosome disorder, genetic variation, exercise therapy

## Abstract

Intellectual disability (ID) is a prevalent neurodevelopmental disorder characterized by neurodevelopmental defects such as the congenital impairment of intellectual function and restricted adaptive behavior. However, genetic studies have been significantly hindered by the extreme clinical and genetic heterogeneity of the subjects under investigation. With the development of gene sequencing technologies, more genetic variations have been discovered, assisting efforts in ID identification and treatment. In this review, the physiological basis of gene variations in ID is systematically explained, the diagnosis and therapy of ID is comprehensively described, and the potential of genetic therapies and exercise therapy in the rehabilitation of individuals with intellectual disabilities are highlighted, offering new perspectives for treatment approaches.

## 1. Introduction

Intellectual disability (ID) is a common trait often accompanied by intellectual dysfunction and other neurological conditions, with specific functional deterioration occurring at different developmental stages. Intellectual function refers to a person’s ability to understand the external world and participate in activities. Individuals with ID often have difficulty comprehending new or complex information, acquiring new skills, and coping with external situations. These symptoms typically arise before adulthood and are often associated with neurological disorders such as Coffin–Siris syndrome, Rett syndrome, Gerstmann syndrome, Trisomy 21 syndrome and so on. ID and autism spectrum disorder (ASD) share some similar impairments in social communication, daily behavior, and language development, but there are significant differences. For instance, the abnormal phenotypes of ID are generally associated with broader brain structural and microscopic abnormalities, whereas there are no such associations in ASD. ID is characterized by significantly below-average intellectual functioning, and certain forms of ID have distinct genetic patterns and “ID genes” which are not present in ASD. Therefore, unlike ASD, the diagnosis of and research into ID has focused more on genetic abnormalities. The genetic causes of ID include chromosomal abnormalities, epigenetic defects, single gene variations, multiple gene abnormalities, copy number variations, and non-coding RNA abnormalities, which can lead to congenital metabolic errors and neurodevelopmental defects [1].

Based on the presence of comorbidities and clinical features [2], ID can be classified as either syndromic (S-ID) or non-syndromic (NS-ID). ID can be classified as either syndromic (S-ID) or non-syndromic (NS-ID) based on the presence of comorbidities and clinical features. Traditionally, NS-ID patients present with ID as the sole clinical feature. However, due to the difficulty of excluding subtle neurological abnormalities, NS-ID is challenging to identify, leading to a blurred boundary between the two categories. Consequently, S-ID and NS-ID do not have a clear-cut relationship.

In terms of severity, ID can be classified into four levels based on IQ test results, with an IQ of <20 classified as profound, 20–35 as severe, 36–51 as moderate, and 55–69 as mild ID [3]. An IQ between 70 and 85, although below normal intelligence levels, is not considered ID [4]. Mild ID is often caused by interactions between multiple genes and non-hereditary factors, while severe ID is generally caused by serious sudden events, such as perinatal asphyxia, poisoning, infection, and hereditary factors. Hereditary factors, which include chromosomal abnormalities and single gene variations, are the main causes. Several studies have reported that approximately 50% of known ID cases are related to genetics [4], and that this proportion increases with the severity of ID (even up to 60%) [3,5], suggesting that genetic variations are some of the main predisposing factors.

As gene sequencing technology advances, an increasing number of gene variations associated with ID have been discovered. According to the pattern of inheritance, ID caused by single gene variations can be divided into autosomal dominant ID, autosomal recessive ID, and X-linked ID (XLID). Currently, over 100 genes have been associated with XLID [6], but autosomal mutations are a more common cause of ID. Autosomally inherited ID exhibits strong heterogeneity, with new genes continually being discovered. The discovery of new gene variations plays a crucial role in revealing the pathogenesis and treatment strategies of ID. Previous studies have primarily focused on the general treatment of ID. However, in recent years, genetic treatment has developed rapidly, and appropriate physical exercise has been identified as a means to promote the treatment of ID.

Above all, this review systematically outlines the physiological foundations of genetic variations associated with ID and introduces both genetic therapies and exercise therapy, critically evaluating their therapeutic potential in the treatment of ID and thereby providing new approaches to improve life quality of ID patients.

## 2. Identification

The identification of ID in a clinical setting is approached through either a phenotype-first or genotype-first method. The clinical diagnostic protocol for ID mainly relies on the phenotype-first approach. This method involves a comprehensive clinical history, physical examination and IQ tests, as well as relevant assessments such as electroencephalography, magnetic resonance imaging, and metabolic function tests. These procedures help to identify specific gene variations through the analysis of their clinical manifestations [7]. The clinical history should emphasize health care during the prenatal, perinatal, and postnatal periods and include the results of all previous studies. The physical exam should focus on secondary abnormalities, congenital malformations, somatometric measurements, and evaluations of the neurological and behavioral phenotype. If the results suggest a particular etiological diagnosis, further disease-related analyses are requested; otherwise, metabolic clinical evaluations and high-resolution cytogenetic studies should be conducted [8]. In infancy (0–2 years), delayed gross motor development is predominant, while school-aged children (6–12 years) often show language and learning disorders. ID is frequently accompanied by neurological conditions such as attention deficit disorder, hyperactivity disorder, and autism, necessitating a multidisciplinary treatment approach. However, most ID cases cannot be detected in time; for example, mild ID is often not identified in patients until between the ages of 5 and 9. This delay can be attributed to parents’ reluctance to acknowledge their child’s ID and the psychological burden faced by these children who fear not being accepted by their peers. Furthermore, the symptoms of ID exhibit clinical heterogeneity, and many individuals with mild ID initially appear normal in terms of facial features, motor skills, and understanding, which can also contribute to delayed detection. Due to the psychological burden, the early concealment of non-severe ID and NS-ID, and societal cognitive biases towards ID, patients often cannot receive timely treatment in the early stages, missing the golden treatment period and causing significant pressure for later treatment.

In addition to conducting clinical phenotypic assessments, a heredofamilial history spanning at least three generations should be obtained to investigate family antecedents of ID and fragile X syndrome (FXS), and FXS gene testing should be performed for all patients. Table 1 shows the characteristics of different gene sequencing technologies. During the latter half of the 20th century, Sanger sequencing was the sole method of genetic variation testing available, with a sequencing length of up to 1000 bp and an accuracy rate of almost 100%. However, due to its labor-intensive nature and the fact that only one gene could be tested at a time, it was both time consuming and expensive. Additionally, the phenotype-first approach has limited effectiveness for patients with multiple comorbidities, ID of unknown genetic origin, and pathologies with unknown mutated genes. As a result, second-generation sequencing technologies have emerged that do not require known genomic loci.

Next-generation sequencing (NGS), also called high-throughput sequencing technology and including whole-exome sequencing (WES) and whole-genome sequencing (WGS), emerged in the early 21st century [9] and has been applied to many species [10]. WES can efficiently identify pathogenic variations because exons are the protein-coding regions of genes where most known variations occur. WES has a diagnostic rate of 30% in neurodevelopmental conditions, including ID, and is often preferred in developed countries due to the short turnaround time, low cost, and high diagnostic rate. However, DNA sequences outside of the exonic regions may also affect genes, leading to ID and other conditions. Therefore, WGS is important for identifying variations in other parts of the genome, although it is more expensive than WES. Long-read WGS can interrogate more than 10,000 bp of a genomic DNA sequence [11] and identify pathogenic variations in complex genes, even those not fully resolved by prior evaluations [12], improving the detection of de novo variations and increasing the recognition of variations in neurodevelopmental disorders, including ID, which is not possible with short-read WGS. Long-read WGS is part of third-generation sequencing (TGS), which can detect novel transcripts from RNA-seq and methylation status using products from Oxford Nanopore Technologies (ONT, Oxford, United Kingdom), thus providing more information about disease-associated variations [13]. Following improvements in SMRT technology in recent years the PacBio RS sequencer was developed, which is a device that has made significant progress and can read longer lengths and improve overall accuracy. However, this instrument is expensive and has limited throughput per SMRT cell. In addition to diagnostics and discovery, NGS can be used to detect methylation status, alternative splicing, small RNAs, allele-specific expression, and even haplotypes and rearrangements [14].

TGS is a single-molecule sequencing technique that can complete gene sequencing without the need for PCR amplification [15]. TGS overcomes issues related to biases introduced by PCR amplification and dephasing and can significantly increase the length of base reads, even surpassing the results that Sanger sequencing can achieve [16]. Additionally, TGS improves speed and leads to more homogenous genome coverage. However, its high error rate during sequencing has limited its widespread adoption.

Furthermore, non-sequencing technologies have been implemented in practice, including optical genome mapping (OGM). OGM has the ability to detect genome structural variations at a high resolution, identify new gene variations [17], and is considered a molecular substitute for conventional cytogenetic testing [18].

## 3. Physiological Basis

Gene variations will lead to transcriptional dysregulation, signaling pathway abnormalities and abnormal protein expression, and impairing neural and synaptic functions. Synaptic impairments and neuronal dysfunction are common in neurodevelopmental disorders that include ID symptoms, involving reduced synaptic plasticity, impaired synaptic maturation and neuronal development abnormalities. The function of learning and memory relies on the plasticity of synapses. Impairments in synaptic plasticity can decrease synaptic connections, leading to reduced synaptic transmission, affecting cognitive and learning abilities, and resulting in ID. The morphology of neurons is crucial for synaptic transmission and plasticity. Neurons consist of dendrites and axons, which transmit and process information. Dysfunction of these structures can result in cognitive impairments, and it is common to observe pathological features of dendritic spines and axons in patients with ID. The dendritic spines, where most excitatory synapses are located, are often affected in ID. FXS is the most common disease to cause dendritic spine damage. Pyronneau et al. dissected the neural tissues of FXS patients and found a higher proportion of immature dendritic spines, leading to synaptic dysfunction and cognitive impairment [19]. A morphological analysis of neurons in mice with ID caused by *PRR12* variations revealed a decrease in dendrite number and length compared with normal mice. Furthermore, Golgi staining of hippocampal and cortical pyramidal neurons in patients with Gerstmann syndrome showed a decrease in the dendritic number and branching compared with individuals without the syndrome. Additionally, induced pluripotent stem cells (iPSCs) can generate functional neurons; however, Rett syndrome patient-derived iPSC neurons have fewer synapses than controls [20] (Figure 1).

Transcriptional dysregulation can result in the abnormal protein synthesis of regulatory genes, leading to abnormal protein expression, gene variations, and ultimately ID. The TCF4 transcription factor is highly abundant in the neurons, and its transcriptional defects can cause the dysregulation of the central nervous system’s language and memory functions, causing severe ID [21,22]. YY1 is a zinc-finger transcription factor, the decrease in which will lead to dysfunction of the brain and nerve development. Michele Gabriele et al. have confirmed YY1’s role in cognitive impairment and ID by immunoprecipitating YY1-bound chromatins and identifying the protein at both ends with antibodies [23].

Abnormalities in cell signaling pathways are significant causative factors in ID. For instance, variations in the RAS-MAPK pathway can disrupt the normal functioning of the MAPK signaling cascade, leading to the dysregulation of growth factors and embryonic development metabolism [24] and resulting in delayed neurodevelopment and ID [25]. Research has demonstrated that the dysregulation of Rho GTPase can result in abnormalities in the ARHGEF6/Rac-Cdc42/PAK3/LIMK1 pathway [26], leading to irregularities in neuronal plasticity and spinal morphology, which can cause neurodevelopmental disorders and neurodegenerative diseases. The Wnt signaling pathway is mainly involved in the formation of neuronal axons, dendrites, and synapse development through both the β-catenin signaling pathway and β-catenin-independent signaling pathways, the impact of which will result in ID [27]. Some gene mutations will lead to calcium channelopathies, which can cause cerebellar morphological changes and lead to several neurological disorders, including ID [28,29].

Proteins play a crucial role in the development and function of the nervous system, and their abnormal expression is closely linked to ID. De novo variations in the autosomal gene *SYNGAP1*, which truncate the protein, have been identified in approximately 3% of NS-ID patients. These variations lead to decreased synaptic plasticity, impacting the patient’s cognitive and learning abilities [30]. Arid1a haploinsufficiency can cause a decrease in the dendritic spine number and branching in mice, further reducing synaptic variability and leading to a decrease in the content of long-term potentiation (LTP) in the postsynaptic density (PSD) of hippocampal tissue. This can result in Coffin–Siris syndrome [31]. In animal experiments, a reduction in the expression of CDKL5 protein in excitatory neurons in rats was found to inhibit dendritic growth and reduce dendritic and spine branching, leading to ID and triggering Rett syndrome [21].

However, there are numerous gene mutations that cause ID and we present the most common of these in Table 2.

## 4. Genetic Etiology of ID

Genetic causes of ID include monogenic disorders, polygenic disorders, and chromosomal aberrations, with the latter being identifiable in up to 25% of individuals with ID [32]. Polygenic disorders have a heritability of 70–80%, with genetic factors playing a predominant role and demonstrating familial aggregation. The involvements of numerous mutated and susceptibility genes contribute to the complexity of the etiology. Consequently, polygenic disorders have emerged as a focal point in international disease genomics research. Trisomy 21 syndrome is the most common chromosomal disorder leading to ID, primarily causing developmental differences in cognition and learning. 

In patients with Down’s syndrome (DS), some neurons in the brain transform into astrocytes, and an excess of glial cells can reduce synaptic density and cause dendritic spine dysfunction, hindering dendritic development [33]. In DS patients, the high activity of the DYRK1A protein kinase is inhibited, leading to the overexpression of the DYRK1A protein, which affects neuronal plasticity, resulting in cognitive abnormalities and memory impairment [34]. In addition, abnormal neurotransmitter transmission is a major factor contributing to the clinical symptoms of DS. Excessive inhibition and dysfunction of GABA in the brains of DS patients affects the balance between excitatory and inhibitory synapses, disrupting synaptic plasticity and leading to learning difficulties, cognitive dysfunction, and memory disorders [35]. The loss of other neurotransmitters, such as serotonin (5-HT) [36], noradrenergic, and histaminergic [37], is also associated with cognitive and learning disabilities in DS. A damaged neuronal structure and function, as well as metabolic abnormalities, are important factors in DS-induced ID. As DS is a comprehensive ID disease, many underlying mechanisms remain to be elucidated.

Structural changes classify chromosomal aberrations into numerical and partial chromosome abnormalities, such as polyploidy, monosomy, microdeletion, and microduplication syndromes [38]. Reports have shown that approximately 2% of unexplained ID cases are diagnosed as being due to either chromosomal microdeletions or microduplications [39]. Among these ID-causing variations, Angelman syndrome (AS) is relatively common and is often associated with severe cognitive dysfunction, speech disorder, and dyskinesia. *UBE3A* can regulate synaptic function and plasticity [40], and recurrent deletions of the maternal chromosome 15 (15q11. 2-q12) result in the loss of *UBE3A* function, inhibiting the degradation of ubiquitinated substrate proteins and causing neuronal dysfunction [41]. Studies in maternal *UBE3A* deletion and *UBE3A* knock-in mice have shown that, although the brain morphology of the mice is normal, synaptic plasticity in the neocortex is impaired, and the length and density of the hippocampal dendritic spines are reduced [42,43], which may be one of the reasons for impaired neuronal function. However, the mechanism by which *UBE3A* regulates synaptic function remains to be elucidated. When the microdeletion of chromosome 15 occurs on the paternal chromosome, Prader–Willi syndrome results, which has a less severe degree of ID than AS [4].

According to the pattern of inheritance, single-gene variations are categorized as autosomal dominant ID, autosomal recessive ID, and X-linked ID (XLID).

### 4.1. Autosomal Dominant ID

Over 400 genes have been associated with autosomal dominant ID [44]. Mild autosomal dominant ID typically carries one copy of a mutant allele and one normal allele on a gene7, with a degree of familial inheritance. Moderate and severe autosomal dominant ID conditions are mainly reported with de novo variations, rarely replicate genetically, and mainly manifest as de novo variations [4], often including spontaneous chromosomal deletions. The pathogenic genes are usually identified by mapping the mutated chromosomes. Autosomal dominant monogenic diseases usually cause mild ID, characterized by learning disabilities [45]. Spontaneous deletions lead to haploinsufficiency, as in neurofibromatosis type I, which is a typical example and has a high prevalence of ID, manifesting mainly as cognitive dysfunction and learning disabilities [46]. The complete loss of the *NF1* gene can result in behavioral abnormalities. The *NF1* gene regulates neural plasticity and neuronal differentiation, including the expression of the GAP43 protein and neurofibromin [47], and affects the number of mature neurons by regulating *cAMP* protein levels [48], thereby participating in early brain development and regulating brain function. This explains why the haploinsufficiency caused by neurofibromatosis type I leads to ID. ID type 5 is related to variations in *SYNGAP1*, and the haploinsufficiency of *SYNGAP1* leads to reduced synaptic plasticity and learning function in mice [49], with different variation sites having different effects on neuronal activity [50]. *SYNGAP1* variations will lead to de novo variations in certain proteins (such as *RASGAP* and *QTRV*), resulting in the loss of domains associated with synaptic plasticity and spine morphology [51] and causing autosomal dominant intellectual developmental disorder-5 (MRD5), which is accompanied by symptoms of moderate to severe ID, such as reduced learning ability.

The *SET* gene encodes a multifunctional protein which is involved in a variety of biological processes, including histone acetylation, transcriptional control, chromatin remodeling, cell migration, apoptosis, and cell cycle regulation [52]. This gene is widely expressed in the human body and plays an important role in various organs, such as the kidney, liver, and brain. In 2017, research suggested that abnormalities in *SET* could downregulate *CTNNB1*, which may be associated with X-linked intellectual development disorder-19 [2]. Subsequently, Servi J. C. Steven and colleagues used WES to identify six cases of ID caused by de novo variations in the *SET* gene [53]. In 2023, Pan X and other researchers identified two unrelated ID patients with an *SET* gene variation in China [54]. Through a combination of genetic sequencing with clinical symptoms, they discovered that a haploinsufficiency of *SET* in the human body can lead to brain developmental and functional deficits, causing non-syndromic ID. NS-ID has been associated with various developmental delays and other clinical features in areas such as development, motor skills, and language. At the same time, some researchers have also found that the overexpression of the SET protein can lead to ID in individuals with Alzheimer’s disease, and other researchers have also demonstrated that the low expression or knockout of the SET protein in the body can lead to the abnormal function of these proteins, resulting in embryonic lethality [55]. However, whether variations in the *SET* gene definitively cause ID and the underlying mechanisms of its effects require further exploration.

### 4.2. Autosomal Recessive ID

Autosomal recessive ID is the most common genetic cause of ID [53], accounting for a quarter of all inherited ID cases [4]. Individuals with autosomal recessive ID will have inherited one defective gene from each parent who is a carrier without symptoms. A typical autosomal recessive genetic disorder is phenylketonuria (PKU), an inherited metabolic disorder in which the *PAH* gene mutation on chromosome 12 leads to a phenylalanine metabolism disorder, causing a range of physical disorders. Elevated levels of phenylalanine in the body result in reduced dendritic arborization and fewer synaptic connections in neurons [56] and abnormal glucose metabolism in the brain, leading to cognitive impairment and severe memory disorder [57]. In addition, the deficiency of monoamine neurotransmitters caused by high phenylalanine levels, such as 5-HT and norepinephrine [58], also leads to cognitive and behavioral differences [59]. Even with strict dietary control from early childhood, patients still have a higher probability of developing attention deficit hyperactivity disorder (ADHD) and learning disabilities than the general population [60,61]. *TRAPPC9* deficiency affects nerve growth factor-induced neuronal differentiation and influences synaptic plasticity [62], regulating adult neurogenesis. Existing research suggests that loss-of-function (LOF) mutations in the *TRAPPC9* gene can cause moderate to severe ID and highly specific brain abnormalities [63]. NSUN2 protein affects the translational regulation of synaptic plasticity [64], and an analysis of 16 cases found that the *NSUN2* variations causing autosomal recessive ID are all moderate to severe, possibly due to nonsense-mediated mRNA decay [65].

### 4.3. XLID

The X chromosome contains many genes relating to learning and cognition, so partial damage to the X chromosome can lead to ID. In the early 1920s, researchers proposed XLID, which was subsequently confirmed in those families with X-linked inheritance patterns [66]. XLID accounts for 10–20% of inherited ID cases [67]. In most families with an X-linked inheritance pattern, females are often carriers and pass on the mutated gene to the next generation, which typically affects only males. However, due to the complete variation in *FMR1* in males, who are almost invariably infertile, there are some ID conditions with a higher prevalence in female patients, such as FXS [68]. FXS is the most common monogenic genetic disorder leading to ID, accounting for 5% of all cases [69]. In a similar manner to Huntington’s disease (HD), FXS is caused by the expansion of a trinucleotide (CGG) repeat in the *FMR1* gene located at the distal end of the long arm of the X chromosome (Xq27. 3). The inheritance pattern of FXS does not follow classic Mendelian laws but mainly depends on the number of CGG repeats in the *FMR1* gene; the larger the number of repeats, the more severe the ID. Based on the number of CGG expansions, mutations can be categorized into premutation (55–200 repetitions) and full mutation (>200 repetitions) [70]. The greater the number of CGG repeats, the more severe the symptoms. A premutation, which is usually associated with milder symptoms or an increased risk of other related conditions, has a higher likelihood of expanding into a full mutation when passed to the subsequent generations. In subsequent generations, the premutation often results in milder symptoms or an increased risk of related conditions, but does not manifest as the full-blown disease. Conversely, a full mutation can directly induce symptoms and is associated with the genetic disorder. One study found that the trinucleotide repeats in mothers of FXS children expand from premutation to full mutation; previously, mothers were typically considered asymptomatic carriers until their child was diagnosed [71]. FXS is the most common genetic cause of ID after Down’s syndrome, accounting for 5% of ID patients [7]. Approximately 80% of males and 70% of females with FXS have ID [72], with cognitive impairment generally being more severe in males. The *FMR1* gene produces the fragile X messenger ribonucleoprotein (FMRNP), which is essential for brain development, with its transcriptional disruption leading to reduced FMRNP levels [73]. The absence of FMRNP stimulates the metabotropic glutamate receptor type I, which inhibits signaling in neuronal dendrites and synapse maturation [74]. In addition, FXS often leads to a reduction in the volume of the posterior vermis of the cerebellum, resulting in a lower level of IQ and causing ID [75].

Rett syndrome is considered fatal for male fetuses, leaving only female patients [45], and is caused by *MECP2* deficiency. *MECP2* variations affect functions in different types of cells and different regions of the brain [76]. As a result, *MECP2* variation causes not only cognitive dysfunction, speech disorder, and communication problems, but also breathing difficulties and a host of other issues [77]. *MECP2* is widely distributed in the human body, with the highest concentration in the brain, where it can repress the transcription of certain non-essential genes. Abnormal deficiencies in *MECP2* can lead to the overexpression of certain genes in the brain, affecting the maturation of the nervous system [78]. *MECP2* plays a critical role in maintaining the normal morphology of neuronal dendrites, and variations in and deficiency of *MECP2* affect synaptogenesis and plasticity, reducing both neuronal dendrite complexity and brain volume [79], affecting neuronal function and ultimately leading to learning disabilities and cognitive impairment.

## 5. Treatment

### 5.1. General Treatment

A substantial body of research has indicated that the brain’s plasticity is most pronounced before the age of 5, a developmental period that also offers the most significant potential for the treatment of neurodevelopmental disabilities and the reconstruction of brain functions. Therefore, it is of paramount importance that ID patients commence treatment as early as possible. The differences observed in ID patients are multifaceted, encompassing impairments in social communication, language abilities, learning skills, and cognitive functions. Consequently, the treatment objective should be to normalize daily life, develop a rehabilitation plan with multidisciplinary cooperation, and provide personalized treatment plans for patients.

Firstly, appropriate education should be provided to patients. For ID patients without hearing or vision disabilities, a minimally restrictive learning environment can effectively improve their academic performance. Special education for individuals with ID should not be limited to knowledge teaching but should also focus on how to communicate with others, improving vocational skills and communication abilities, and training in functional skills (such as dressing and washing), enabling people with ID to live as independently as possible and reducing the need for external support. Due to neurodevelopmental disabilities, people with ID may sometimes exhibit behavioral and mental abnormalities, such as self-harm, violent tendencies, obsessive–compulsive disorder, and ASD. People with ID should be provided with appropriate assistance in addressing possible behavioral issues (such as grounding), while positive reinforcement should be provided for appropriate behaviors (such as praise). Concurrently, it is essential to work alongside people with ID in identifying and addressing negative thoughts, alleviating mental stress and thereby modifying negative behaviors.

Pharmacotherapy represents an essential component of clinical treatment; however, its primary objective is to address the comorbidities associated with ID, such as autism, attention deficits, and hyperactivity, rather than ID itself. In clinical trials, risperidone and aripiprazole have been demonstrated as efficacious in the treatment of violent tendencies [80] and ASD [81] caused by childhood ID, and these drugs have certain safety profiles that have been approved by the U.S. Food and Drug Administration for clinical use [82]. However, due to the significant side effects of antipsychotic medications, there is still a lack of evidence on the optimal dosage for many psychotropic drugs. Currently, the Dyskinesia Identification System: Condensed User Scale (DISCUS) and the Matson Evaluation of Drug Side Effects (MEDS) are the most commonly employed instruments in clinical settings for evaluating the efficacy and side effects of medications [83]. These instruments serve to guide the subsequent steps in medication use.

### 5.2. Genetic Therapy

Genetic therapies can treat diseases by using viral vector or nanoparticles for gene delivery to introduce, repair, or replace defective or missing genes, providing new ideas and possibilities for solving some difficult-to-treat genetic diseases. IDs are commonly associated with severe and intractable epilepsy; however, the use of anti-epileptic drugs (AEDs) is often not sufficient to prevent developmental delay and cognitive impairments. Gene therapy provides a possible means of preventing ID and epilepsy by restoring correct protein function. Technology such as gene editing and tools such as antisense oligonucleotides (ASOs) have had successes in some clinical trials.

Gene replacement has been shown to be beneficial in treating ID. Various viral vectors have been used in gene delivery, including those derived from adeno-associated virus (AAV), retrovirus, adenovirus (Ad) and herpes simplex virus (HSV). Researchers often choose AAV for gene delivery due to its neurotropism, long-term effects and safety profile. For example, recombinant AAV2/5 vectors have been injected to drive the expression of FMRP in the hippocampus of adult fragile X mice, resulting in the restoration of synaptic function and suggesting the potential to improve cognitive function in fragile X syndrome (FXS) [84].

Gene-editing technology can precisely change DNA sequences at target sites to modify the genome, and this technology has been used in many fields including agriculture, biological sciences, and medicine. In recent years, gene editing has been applied to construct animal models to promote mechanistic studies and human disease treatments, and significant progress has been achieved in cancer, neurological disabilities, cardiovascular disease, regeneration, etc. [85]. There are various gene-editing technologies; among them, *CRISPR-Cas9* is the most common editor providing potential treatment options for ID. The genetic treatment of FXS is an example. CRISPR–Cas9 nuclease and single-guide RNA (sgRNA) can be targeted at the induced iPSCs of patients with FXS in the *FRX* gene upstream of the CGH repeat, resulting in the reactivation of *FMR1*. Furthermore, the recombination of two double-stranded breaks can delete the trinucleotide CGG repeat, which can also increase the levels of *FMR1*, restoring the morphology and function of the brain neurons [86]. Some therapeutic targets aimed at improving ID via restoration of the protein synthesis and synaptic function have shown a certain degree of effectiveness [87]. However, due to developed tolerance, the possibility of failure remains, and where drugs have failed, the introduction of gene-editing technology can raise the potential success rate [88]. Although gene editing may provide a new way of treating ID, the issue of biosecurity should be noted [89], and the regulatory and methodological problems that arise from gene editing are still not fully understood. Gene editing still requires significant improvements to be made before being applied to treat diseases.

As a tool that can selectively reduce the expression of target genes, antisense oligonucleotides (ASOs) have become increasingly important in gene therapy. ASOs are chemically synthesized oligonucleotides, 10–30 nucleotides in length, that bind RNA using Watson–Crick base pairing rules [90]. In the clinical sector, ASOs have been introduced to many diseases, such as spinal muscular atrophy, familial hypercholesterolemia, and Duchenne muscular dystrophy. The success of these treatments has demonstrated the possible applications of ASOs for the treatment of neurodevelopmental disabilities, including ID. For example, some researchers have used ASOs to reactivate the expression of the paternal *UBE3A* allele in AS mice to control UBE3A protein expression, and they discovered that motor coordination deficiency can be significantly improved but not fully rescued and AS neurocognitive phenotypes can be partially rescued [91,92]. Though the ASO treatment was very effective at a molecular level, due to the differences between humans and mice, the insufficient understanding of underlying brain circuits, and the economic burdens of long-term treatment, there is still a need to carefully consider its feasibility and benefits.

### 5.3. Exercise Therapy

It is irrefutable that exercise has a beneficial effect on the average person. Aerobic exercise in particular not only promotes blood circulation and enhances metabolism but also improves the function of organs, effectively enhancing physiological functions. Furthermore, exercise promotes normal brain development and plasticity, enhancing brain function. Studies have demonstrated that mice engaged in exercise exhibit greater total dendritic length in Purkinje cells, higher levels of dendritic spine density [93], and increases in total brain volume and the number of whole-brain fibers compared with sedentary mice [94]. One of the most common forms of aerobic exercise, running, has been shown to effectively increase the density of dendritic spines in the hippocampus, the number of synapses, and the quantity of astrocytes [95]. Long-term runners exhibit a significant reduction in impulsive and aggressive behaviors [96] and improvements in memory and social skills [97]. In addition, other aerobic exercises also improve brain function. For instance, research by Stefan Schneider et al. [98] has proven that cycling has a positive effect on neurogenesis, attention, cognitive functions, and neural plasticity in school-aged children, with long-lasting effects. Additionally, aerobic exercise can increase blood flow to the hippocampus [99], improve cognitive and learning networks [100], and may contribute to the development of self-regulatory neural circuits’ structure and function, further enhancing neural plasticity. Exercise can increase the volume of the hippocampus and the basal ganglia [101], promoting their connection with white matter, thereby improving attention and alleviating emotions with a negative impact such as anxiety, those arising from ASD, and depression [102].

Given the role of exercise in promoting neural development and brain function in healthy individuals, it is reasonable to speculate that exercise plays an important role in the physical fitness and rehabilitation of people with ID (Figure 2). Patients with ID generally have a higher risk of cardiovascular diseases, resulting in an elevated premature mortality rate [103]. A substantial body of research has demonstrated that moderate exercise benefits the cardiovascular and muscular health of adults with mild to moderate ID. It can alleviate and prevent diseases such as skeletal muscle dysfunction, congenital heart disease, and muscle tone decline caused by ID [104], thereby improving patients’ physical fitness. Because people with ID frequently exhibit balance disorders and gait abnormalities, falls represent a significant risk factor for this population [105]. Exercise can enhance neural control over muscles and increase muscle power, making low-intensity walking exercises that exert less pressure on the knee joints a preferable option. In addition to traditional sports, virtual reality (VR)-based exercise or videogaming such as Nintendo Wii Fit can effectively enhance physical function. This form of exercise is enjoyable, highly adherent, and its intensity is comparable to moderate-intensity exercise, making it well-suited to meet the daily activity needs of individuals with ID. Many researchers have reported that VR-based exercise can significantly improve the muscle strength, endurance capacity, and the balance of individuals with ID [106].

Additionally, moderate exercise has been shown to improve cognitive performance in individuals with ID [107]. Alterations in electroencephalogram (EEG) activity in the frontal lobe area are acknowledged for their association with emotional and cognitive abilities. Exercise can reduce cortical current density in the frontal lobes, resulting in increased self-esteem and positive emotions in individuals with ID [108], as well as improvements in social perception and self-awareness [109]. Research by Tobias Vogt and others has shown that appropriate-intensity cycling can affect cortical electrical activity patterns in individuals with ID, temporarily enhancing neuronal activity related to cognition and improving reaction time and decision-making processes [109]. Furthermore, the results of Vogt’s study indicate that regular running training led to increased positive emotions and social acceptance in people with ID [110]. Effective exercise can prevent and alleviate common anxiety symptoms in people with ID, improving their psychological state, enhancing their ability to integrate into social life, and even improving the transmission of brain neurotransmitters, reducing functional impairment [111]. Multiple studies have indicated that short-duration exercise is more effective in improving ID than long-duration exercise [112], and that exercise of low to medium intensity is more effective than high-intensity exercise at improving brain-derived neurotrophic factor (BDNF) levels in the body, thereby enhancing neural plasticity [107].

Undoubtedly, appropriate exercise and related behaviors can benefit individuals with ID. Numerous indicators suggest that physical exercise plays a significant role in enhancing overall functioning and recovering brain function in individuals with ID. Therefore, we recommend that ID patients engage in moderate-intensity full-body aerobic exercise, such as running, dancing, swimming and Taichi [106], to improve physical function. However, further research is required to determine the optimal duration and most suitable intensity of exercise for improving the cognitive symptoms of ID. However, at present, due to factors such as transportation, financial constraints, and a lack of awareness, exercise therapy has not been widely implemented in the treatment of people with ID.

## 6. Conclusions and Outlook

As molecular detection and gene sequencing technologies advance, an increasing number of variations have been discovered, leading to a deeper understanding of the genetic factors associated with ID. Although the clinical diagnosis of ID through patient phenotypes and IQ tests is highly effective, the genetic heterogeneity of ID requires genome-wide approaches. This review categorizes gene mutations leading to ID based on inheritance patterns. Regardless of the specific type of gene mutation, the underlying mechanism typically involves the impairment of neural and synaptic functions. Consequently, current therapeutic research increasingly focuses on targeting the associated genes or proteins, aiming to restore their normal function or expression to recover neuronal function and improve outcomes in ID. Current research on ID primarily focuses on single-gene variations and chromosomal variations, with relatively less research on multi-gene variations. This is likely due to the strong genetic heterogeneity, high lethality, and complex etiology associated with multi-gene variations. A considerable period of time will be required to identify all the gene variations that cause ID. Fortunately, doctors pay more attention to the precision phenotyping method for identifying ID when performing clinical examinations, which enables them to identify any abnormalities in time. In addition, the increasing application of the precision genotyping method, including genetic assessment and sequencing technologies, has effectively reduced the rate of neonatal abnormalities, identified causes for effective treatment, and improved prognostic outcomes. The combination of these factors enables the introduction of an era of precision medicine.

Indeed, the significance of prevention for ID significantly outweighs that of treatment. For example, improvements in public awareness of the negative effects of consanguinity are important. Pregnant women can regularly attend prenatal check-ups to screen for early-detectable gene variations and chromosomal abnormalities, allowing parents to make an informed choice about their affected baby. Following birth, it is advisable for newborns to undergo regular metabolic screening to promptly detect metabolic disorders such as phenylketonuria, ensuring the implementation of appropriate preventative and therapeutic measures. In daily life, attention should also be paid to the child’s behavior and facial features, with medical advice sought if any differences are observed. The lifetime costs of caring for an ID patient are extremely high, yet moderate exercise is a highly cost-effective treatment strategy. Motor skills are fundamental to overall health and quality of life, and exercise therapy should be incorporated into mainstream ID treatment protocols. In daily training, while functional exercises are undertaken to enhance the gross and fine motor skills of individuals with ID, it is advisable to include appropriate moderate-intensity aerobic exercises.

Due to the advances in medical science and technology, there has been a significant reduction in the prevalence of developmental disabilities. Nonetheless, in certain geographic regions, such as tech valleys, the incidence of polygenic developmental disorders persists at elevated levels. This enduring prevalence may be attributed to their potential long-term exposure to environments with heightened radiation levels and the elevated cost of living. The existence of individuals in these regions with ID in excess of national levels appears to contradict the principles of natural selection. Biologically, this phenomenon could be linked to random mutations and other non-selective evolutionary pressures that defy natural selection mechanisms. From a philosophical perspective, development proceeds in a spiraling upward trajectory. The intricate tapestry of biological evolution suggests that the attributes associated with certain differences, while seemingly detrimental, may confer unforeseen advantages under specific ecological conditions. This paradox invites a deeper contemplation of the multifaceted implications of such conditions within the broader context of evolutionary biology and the philosophy of science.

## Figures and Tables

**Figure 1 genes-15-01118-f001:**
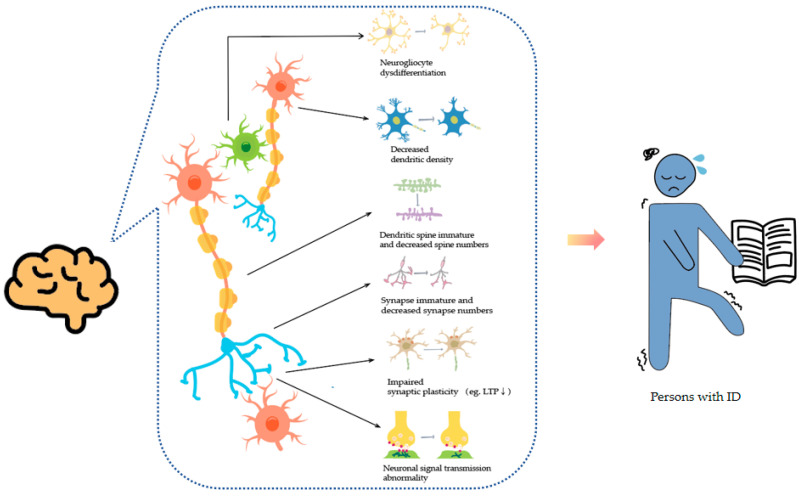
The physiological basis of intellectual disability (ID). Gene mutations related to ID often lead to neurological dysfunction, which in turn can cause ID. Individuals with ID typically exhibit a range of challenges, including cognitive impairments, speech and language difficulties, reading disabilities, motor skill delays, and emotional disturbances. LTP, Long-Term Potentiation. ↓, decrease.

**Figure 2 genes-15-01118-f002:**
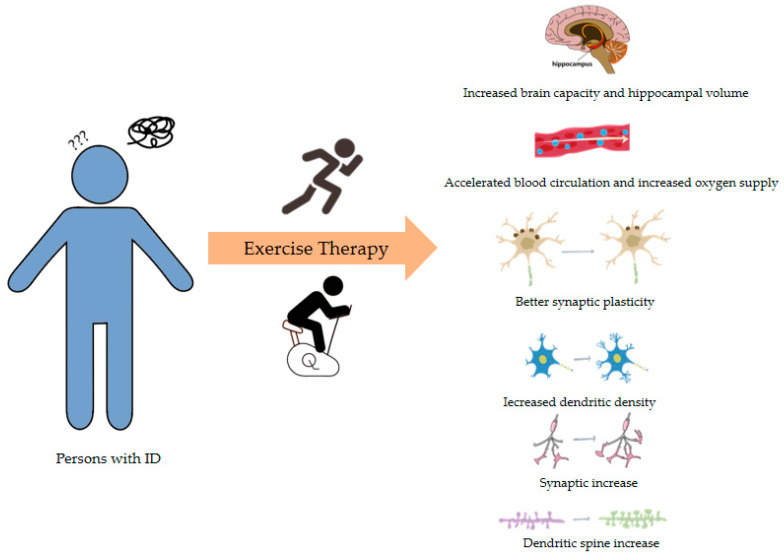
The role of exercise in the physical fitness and rehabilitation of individuals with ID.

**Table 1 genes-15-01118-t001:** The advantages and disadvantages of different gene sequencing technologies.

Name	Catalog	Advantages	Disadvantages
Sanger sequencing		A long sequencing length (1 kb) and a high accuracy rate of 100%	Time consuming (weeks or even months of work) and expensive (millions of dollars), requires known genomic loc
Next-generation Sequencing (NGS)	Whole-exome sequencing (WES)	Short turnaround time (only days or weeks of work), low cost (thousands of dollars), and high diagnostic rate (millions of reads)	Can only search DNA sequences in the exonic regions
	Whole-genome sequencing (WGS)	Captures both exonic and intronic regions, identifies pathogenic variations in complex genes, and improves de novo mutations detection	Short reads (400 bp or 2 × 300 bp)
Third-generation sequencing (TGS)		Overcomes the biases-related issues introduced by PCR amplification and dephasing and increases the length of base reads	High error rate during sequencing (an error rate of 10–20%)

**Table 2 genes-15-01118-t002:** The important pathway and genes that cause intellectual disability (ID).

Gene	Pathogenesis	Phenotype
*MECP2*	abnormal transcriptional regulation and chromatin remodeling	Immature and decreased synapse numbers Impaired synaptic plasticity (e.g., LTP ↓) Decreased dendritic density Dendritic spine immature and decreased spine numbers
*UBE3A*	affecting the ubiquitin-proteasome pathway	
*DYRK1A*	abnormal encode a kinase involved in brain development	
*PTEN*	dysregulated cell growth and proliferation	
*TCE4*	abnormal transcriptional regulation	
*GNAS*	abnormal G protein signaling	
*CHD8*	disrupt chromatin remodeling and gene expression	
*KDM5C*	affect a neurodevelopmental kinase	
*PHF*	neurodevelopmental gene expression dysregulation	
*EHMT1*	neurodevelopmental gene expression dysregulation	
*NF1*	dysregulated cell growth and proliferation	
*ZDHHC9*	abnormal protein palmitoylation	
*ATRX*	chromosomal abnormalities and defective DNA repair	
*FOXP1*	abnormal transcriptional regulation	
*FMR1*	impaired FMRP protein production	
*TSC1/TSC2*	abnormal cell growth and proliferation	
*CDKL5*	affect a neurodevelopmental kinase	
*TRAPPC9*	affects nerve growth factor-induced neuronal differentiation	
*US* *UN2*	decreased tRNA methyltransferase function	
*SHANK3*	abnormal postsynaptic protein density	Impaired synaptic plasticity (e.g., LTP↓) Abnormal neuronal signaling
*SYNGAP1*	de novo variations in certain proteins	
*NRXN1*	abnormal presynaptic protein synthesis	
*SCN2A*	sodium channel dysfunction	
*CACNA1A*	calcium channel dysfunction	
*STXBP1*	dysfunction of neurotransmitter release process	
*GRIN2B*	NMDA receptor dysfunction	
*G* *ABRA1*	GABA receptor dysfunction	

LTP, Long-Term Potentiation. NMDA, N-methyl-D-aspartate. ↓, decrease.

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
