# Peer review of "A 10-Year Review on Advancements in Identifying and Treating Intellectual Disability Caused by Genetic Variations"

_genes, 2024, doi:10.3390/genes15091118_

Round 1

Reviewer 1 Report

Comments and Suggestions for Authors

In the manuscript titled Genetic Variations in Intellectual Disability: Advances in Identification and Treatment, the authors provide informative background on intellectual disability including potential genetic variations and its diagnosis and treatment. This manuscript reviewed the advancement in gene discovery, diagnosis and treatment of intellectual disability and are valuable for the understanding of genetic mutation involved in intellectual disability. However, the structure of this manuscript is poorly organized and it’s hard for readers to follow the key concepts and extract the useful information, which becomes the major flaw of this manuscript. Therefore, major revision must be done.

Major comments:

1.      The title tells the reader that this review paper is focused on advancement, please give a clear time zone on how you chose the papers, from which year to which year. Please also define which is regarded as new information compared to classic discoveries.

2.      In the abstract section, line 14, “the cost of care remains extremely high”, this has nothing to do with the focus of this paper, and you review paper didn’t solve this problem, so this is an inappropriate expression. line 16, “highlights the importance of exercise in its treatment”, from the content of your review paper, there’s not enough information supporting this claim. Only if all the genetic variation are related to motor functions and there are enough experiments data supporting the importance of exercise, you can get this claim. Otherwise, rewrite the abstract to summarize the real aim and real contribution of this review paper.

3.      In the introduction section, line 21-22, the authors need to give a clear definition for intellectual disability (ID), especially relating its characteristics to intellectual functions.

4.      Line 54-55, the summary sentence of introduction section, “emphasizing the introduction of these treatment …, providing new way to …”. This only concluded 1/3 of this paper, which is inappropriate for a review paper. Please rewrite.

5.      What’s the difference between diagnosis and identification? It’s hard to tell the difference between these two sections from the author's description. The author even used “diagnosis” in the first line of the identification section (line 80).

6.      The manuscript used too many different classifications for ID, such as S-ID or NS-ID, mild ID or severe ID, and autosomal ID or XLID, which makes readers confused and hard to follow. Please organize the structure when introducing different classifications or stick to one classification in the whole paper.

7.      In the physiological basis section, there’s obvious overlaps between factors causing ID. The author introduced three factors: cell signaling pathway, cell morphology and transcription dysregulation. These factors are not from the same level, as a result, there will be overlap in its mechanism. Please rewrite or this will cause confusion.

8.      Too many disease names occurred in a sudden, without introducing its relationship with ID. For example, line 148, “Coffin-Siris syndrome”, what’s this? Does it belong to ID? The authors need to provide more background information in introduction part for readers to understand or use a new section to introduce the diseases related to ID.

9.      Too many genes name occurred in each section, the authors need to reorganize the paper, classify these genes into several categories and give sub-title for each category which make reader quickly locate their interested categories for their study.

10. Please always limit your descriptions to ID. Some genes plays a role in the neurodevelopment of brain, it may or may not be a genetic variations of ID. Only when published studies linked this genetic variation to ID, you can report this gene in your review paper. Otherwise, this can cause misunderstandings. For example, in line 176, is there any published study support the role of FoxO in ID?

11. In section 6.3 exercise therapy, the authors introduced the benefits of exercise in heathy people. But less description is given on how the performance of exercise therapy in people with ID. Please rewrite this section and review more papers to support your claim that exercise therapy is effective in patients with ID.

12. The conclusion section didn’t work well to give a summary of the whole paper. The authors spent more than half content in introducing the genetic variations, but the conclusion is focused on exercise treatment, which is not the main focus of the manuscript. Please rewrite.

13. The authors need to give full names for all the abbreviations used for diseases and genes when they first appeared in the manuscript. Please revisit the whole paper and revise.

Minor comments:

1.      Line 5, the title of section should use “diagnosis” instead of “diagnose”, which is a verb.

2.      Too many percentages appeared in the content, such as line 39, 50% and line 187, 25%. The authors should consider using Pie Chart to show these numbers for readers to follow.

3.      The quality of Figure 1 is low, please update. New concepts appeared in the figure, such as “dyslexia”, please add figure legend for readers to understand.

4.      Too many typos existed in the manuscript, for example, Table 1, GGS should be NGS; Line 169, “TTranscriptional”. Please revisit the paper to check and revise them.

5.      Please add references for the introduction section, no reference in the first paragraph. Missing references elsewhere, such as line 135-138.

6.      Line 229-232, “A premutation…”, wrong expression, please revise.

7.      Line 280, what is “autosomal recessive ID”, please specify.

Comments on the Quality of English Language

Some expressions are inappropriate and hard to understand, which needs to be revised. But from the whole perspective, there's no big problem in the quality of English.

Author Response

Response to Reviewers

We appreciate the reviewer’s suggestions, which will help to increase the scientific quality of the paper and improve the resubmitted version. The corrections according to the comments in the revised manuscript are marked in red and underline.

We will answer in a point-to-point fashion to the comments of the reviewer.

Detailed list of changes made to the manuscript, according to comments from Reviewer#1.

Specific feedback

The title tells the reader that this review paper is focused on advancement, please give a clear time zone on how you chose the papers, from which year to which year. Please also define which is regarded as new information compared to classic discoveries.

⇒ Thanks for your comments. Firstly, we mainly select literature from the past ten years as references, and have changed the title into “A 10-Year Review on Advancements in Identifying and Treating Intellectual Disability Caused by Genetic Variations”. Secondly, compared to other reviews, our review introduces both genetic therapies and exercise therapy, critically evaluating their therapeutic potential in the treatment of ID, providing new ways to improve life quality of ID patients. And we also reiterated this new information in abstract and introduction.

In the abstract section, line 14, “the cost of care remains extremely high”, this has nothing to do with the focus of this paper, and you review paper didn’t solve this problem, so this is an inappropriate expression.

⇒ Thanks for your comments. We have deleted it.

line 16, “highlights the importance of exercise in its treatment”, from the content of your review paper, there’s not enough information supporting this claim. Only if all the genetic variation are related to motor functions and there are enough experiments data supporting the importance of exercise, you can get this claim. Otherwise, rewrite the abstract to summarize the real aim and real contribution of this review paper.

⇒ Thanks for your comments. We have changed “highlight the importance of exercise in its treatment” into “highlights the potential of genetic therapies and exercise therapy in the rehabilitation of individuals with intellectual disabilities”. Meanwhile, we have added some content about the effects of exercise on ID patients, making the exercise therapy more convincing.

In the introduction section, line 21-22, the authors need to give a clear definition for intellectual disability (ID), especially relating its characteristics to intellectual functions.

⇒ Thanks for your comments. We have redefined it.

Line 54-55, the summary sentence of introduction section, “emphasizing the introduction of these treatment …, providing new way to …”. This only concluded 1/3 of this paper, which is inappropriate for a review paper. Please rewrite.

⇒ Thanks for your comments. We have rewritten this sentence.

What’s the difference between diagnosis and identification? It’s hard to tell the difference between these two sections from the author's description. The author even used “diagnosis” in the first line of the identification section (line 80).

⇒ Thanks for your comments. The diagnosis section is more focused on clinical practices, while the identification section mainly concentrates on genetics. However, it is challenging to distinguish between the term’s 'diagnosis' and 'identification.' Therefore, we combined the two sections into one: clinical diagnosis is categorized as phenotype-first identification, while genetic sequencing and family history investigation are categorized as gene-first identification.

The manuscript used too many different classifications for ID, such as S-ID or NS-ID, mild ID or severe ID, and autosomal ID or XLID, which makes readers confused and hard to follow. Please organize the structure when introducing different classifications or stick to one classification in the whole paper.

⇒ Thanks for your comments. Other reviews (such as Non-syndromic Intellectual Disability: An Experimental In-Depth Exploration of Inheritance Pattern, Phenotypic Presentation, and Genomic Composition) have also used these classifications, so we retained these categories in our review as well. To easier to follow and avoid confusion, we explained the rationale behind this classification in introduction section.

In the physiological basis section, there’s obvious overlaps between factors causing ID. The author introduced three factors: cell signaling pathway, cell morphology and transcription dysregulation. These factors are not from the same level, as a result, there will be overlap in its mechanism. Please rewrite or this will cause confusion.

⇒ Thanks for your comments. We have phrased this section.

Too many disease names occurred in a sudden, without introducing its relationship with ID. For example, line 148, “Coffin-Siris syndrome”, what’s this? Does it belong to ID? The authors need to provide more background information in introduction part for readers to understand or use a new section to introduce the diseases related to ID.

⇒ Thanks for your comments. We have provided a brief overview of these diseases in the introduction, indicating their relationships with ID.

Too many genes name occurred in each section, the authors need to reorganize the paper, classify these genes into several categories and give sub-title for each category which make reader quickly locate their interested categories for their study.

⇒ Thanks for your comments. We have classified these genes into several categories and give sub-title for each category: 4.1. Autosomal Dominant ID, 4.2. Autosomal Recessive ID, 4.3. XLID.

Please always limit your descriptions to ID. Some genes plays a role in the neurodevelopment of brain, it may or may not be a genetic variations of ID. Only when published studies linked this genetic variation to ID, you can report this gene in your review paper. Otherwise, this can cause misunderstandings. For example, in line 176, is there any published study support the role of FoxO in ID?

⇒ Thanks for your comments. We have deleted genetic variations unrelated to ID.

In section 6.3 exercise therapy, the authors introduced the benefits of exercise in heathy people. But less description is given on how the performance of exercise therapy in people with ID. Please rewrite this section and review more papers to support your claim that exercise therapy is effective in patients with ID.

⇒ Thanks for your comments. We have rewritten this section and review more papers to support my claim.

The conclusion section didn’t work well to give a summary of the whole paper. The authors spent more than half content in introducing the genetic variations, but the conclusion is focused on exercise treatment, which is not the main focus of the manuscript. Please rewrite.

⇒ Thanks for your comments. We have rewritten this section.

The authors need to give full names for all the abbreviations used for diseases and genes when they first appeared in the manuscript. Please revisit the whole paper and revise.

⇒ Thanks for your comments. We have revised them through the manuscript.

Minor comments:

Line 5, the title of section should use “diagnosis” instead of “diagnose”, which is a verb.

⇒ Thanks for your comments. We have revised this section.

Too many percentages appeared in the content, such as line 39, 50% and line 187, 25%. The authors should consider using Pie Chart to show these numbers for readers to follow.

⇒ Thanks for your comments. You are correct. Creating a pie chart would indeed provide clearer visualization. Unfortunately, due to incomplete data, some relevant information on certain types of ID is unclear, so we are unable to create a pie chart.

The quality of Figure 1 is low, please update. New concepts appeared in the figure, such as “dyslexia”, please add figure legend for readers to understand.

⇒ Thanks for your comments. We have redrawn this chart.

Too many typos existed in the manuscript, for example, Table 1, GGS should be NGS; Line 169, “Transcriptional”. Please revisit the paper to check and revise them.

⇒ Thanks for your comments. We have checked and revised them.

Please add references for the introduction section, no reference in the first paragraph. Missing references elsewhere, such as line 135-138.

⇒ Thanks for your comments. We have added them.

Line 229-232, “A premutation…”, wrong expression, please revise.

⇒ Thanks for your comments. Other relevant literature (such as Data-driven phenotype discovery of FMR1 premutation carriers in a population-based sample) has also used this expression, so we retained this form of expression.

Line 280, what is “autosomal recessive ID”, please specify.

⇒ Thanks for your comments. We have specified it.

Some expressions are inappropriate and hard to understand, which needs to be revised. But from the whole perspective, there's no big problem in the quality of English.

⇒ Thanks for your comments. We have used Author Services’ English language editing services of MDPI for editing English language.

Reviewer 2 Report

Comments and Suggestions for Authors

The authors presented a review about the application of genetic variations in the diagnosis and treatment of intellectual disability (ID) patients. A major issue with this manuscript is why the authors highlighted the importance of exercise in its treatment in section 6.3 and Figure 2. It is confusing to readers because this review should be focused on genetic mutation and its association with this disease. Please revise and re-organize the manuscript thoroughly to keep a clear theme of this review. Also, the authors should summarize what genomic mutations have been found to cause ID, in the form of a table or figure.

Minor comments:

1.     In Table 1, Next-generation sequencing(GGS), GGS should be NGS

2.     In Table 1, “Can search DNA sequences outside of the exonic regions”, just say “WGS captures both exonic and intronic regions”

Comments on the Quality of English Language

Extensive editing of English language required.

Author Response

Response to Reviewers

We appreciate the reviewer’s suggestions, which will help to increase the scientific quality of the paper and improve the resubmitted version. The corrections according to the comments in the revised manuscript are marked in red and underline.

We will answer in a point-to-point fashion to the comments of the reviewer.

Detailed list of changes made to the manuscript, according to comments from Reviewer#2.

The authors presented a review about the application of genetic variations in the diagnosis and treatment of intellectual disability (ID) patients. A major issue with this manuscript is why the authors highlighted the importance of exercise in its treatment in section 6.3 and Figure 2. It is confusing to readers because this review should be focused on genetic mutation and its association with this disease. Please revise and re-organize the manuscript thoroughly to keep a clear theme of this review. Also, the authors should summarize what genomic mutations have been found to cause ID, in the form of a table or figure.

⇒ Thanks for your comments. This review systematically outlines the physiological foundations of genetic variations associated with ID, introduces both genetic therapies and exercise therapy, critically evaluating their therapeutic potential in the treatment of ID, thereby providing new approaches to improve life quality of ID patients. According to your comments, we re-organized the manuscript, and deleted some content of exercise therapy.

Also, the authors should summarize what genomic mutations have been found to cause ID, in the form of a table or figure.

⇒ Thanks for your comments. We have summarized what genomic mutations have been found to cause ID in Table2.

Minor comments:

In Table 1, Next-generation sequencing (GGS), GGS should be NGS

⇒ Thanks for your comments. We have revised Table 1.

 In Table 1, “Can search DNA sequences outside of the exonic regions”, just say “WGS captures both exonic and intronic regions”

⇒ Thanks for your comments. We have revised Table 1.

Extensive editing of English language required.

⇒ Thanks for your comments. We have used Author Services’ English language editing services of MDPI for editing English language.

Reviewer 3 Report

Comments and Suggestions for Authors

This manuscript presents a review of ID. It is very thorough, well organized and well written.

A few minor typos can be found throughout, e.g. line 207 should be Angelman syndrome (AS, or line 232 should be “While”—to be corrected throughout.

In Table 1, it might be worth adding the associated costs when you mention “low cost” or “expensive”. In addition, it might be worth specifying error rates e.g. when you mention “high error rate during sequencing”, or specifying the time required when you note “time-consuming”. The more specifics, e.g. in parentheses, the better.

Under Figure 1, it would be worth specifying what you mean by ‘synapse reduction’ (in size? Number?), ‘impaired synaptic plasticity’ (dampened LTP?), and ‘dendritic spine reduction’ (in size? Number?). Again, the more specific, the better.

In addition, might you consider including the different effects on the neuron in the same big picture of a neuron (with arrows pointing to the different areas affected & their associated labels)? Visually this would help readers and scientists generally see the big picture: impaired neuronal network formation, reflected morphologically and functionally.

Relatedly, could you include the whole-person effects on one diagram of one person, with arrows & associated labels? This would again visually prime a more wholistic understanding of the impacts.

Throughout the text, please de-capitalize ‘syndrome’ when used to define a syndrome, e.g. in Table 2 ‘Angelman syndrome’.

Under the “Identification” section you mention the genotype vs phenotype first approaches for the diagnosis of ID. It would be worth mentioning in end of the manuscript (e.g. Conclusion and Outlook section) how there is a movement towards precision phenotyping and precision genotyping, meaning that ultimately (and with enough compute power) we will be able to usher in an era of precision medicine that combines both (ideally). Including one such sentence or two would add context/vision to your manuscript.

Whether in the Conclusion and Outlook section or introductory sections, it would add richness to discuss the differences found (genetically and anatomically/physiologically) between ID and ASD, which in and of itself remains an active area of research of great relevance to many different disorders under study.

Furthermore, it might be fascinating philosophically to include a brief discussion on why the polygenic developmental disorders remain at such high numbers, especially among certain populations (e.g. higher rates of autism in tech valleys). How does it make sense that some persist, counterintuitively from an evolutionary perspective? This would add interesting big picture context to your good review.

Comments on the Quality of English Language

No English editing required; just typos throughout. Please use a spellcheck.

Author Response

Response to Reviewers

We appreciate the reviewer’s suggestions, which will help to increase the scientific quality of the paper and improve the resubmitted version. The corrections according to the comments in the revised manuscript are marked in red and underline.

We will answer in a point-to-point fashion to the comments of the reviewer.

Detailed list of changes made to the manuscript, according to comments from Reviewer#3.

Detailed list of changes made to the manuscript, according to comments from Reviewer#3.

A few minor typos can be found throughout, e.g. line 207 should be Angelman syndrome (AS, or line 232 should be “While”—to be corrected throughout.

⇒ Thanks for your comments. We have corrected throughout.

In Table 1, it might be worth adding the associated costs when you mention “low cost” or “expensive”. In addition, it might be worth specifying error rates e.g. when you mention “high error rate during sequencing”, or specifying the time required when you note “time-consuming”. The more specifics, e.g. in parentheses, the better.

⇒ Thanks for your comments. We have revised Table 1.

Under Figure 1, it would be worth specifying what you mean by ‘synapse reduction’ (in size? Number?), ‘impaired synaptic plasticity’ (dampened LTP?), and ‘dendritic spine reduction’ (in size? Number?). Again, the more specific, the better.

⇒ Thanks for your comments. We have revised Figure 1.

In addition, might you consider including the different effects on the neuron in the same big picture of a neuron (with arrows pointing to the different areas affected & their associated labels)? Visually this would help readers and scientists generally see the big picture: impaired neuronal network formation, reflected morphologically and functionally.

⇒ Thanks for your comments. We have revised Figure 1.

Relatedly, could you include the whole-person effects on one diagram of one person, with arrows & associated labels? This would again visually prime a more wholistic understanding of the impacts.

⇒ Thanks for your comments. We have revised Figure 1.

Throughout the text, please de-capitalize ‘syndrome’ when used to define a syndrome, e.g. in Table 2 ‘Angelman syndrome’.

⇒ Thanks for your comments. We have revised ‘syndrome’ throughout the manuscript.

Under the “Identification” section you mention the genotype vs phenotype first approaches for the diagnosis of ID. It would be worth mentioning in end of the manuscript (e.g. Conclusion and Outlook section) how there is a movement towards precision phenotyping and precision genotyping, meaning that ultimately (and with enough compute power) we will be able to usher in an era of precision medicine that combines both (ideally). Including one such sentence or two would add context/vision to your manuscript.

⇒ Thanks for your comments. We have added the context to our manuscript. 

Whether in the Conclusion and Outlook section or introductory sections, it would add richness to discuss the differences found (genetically and anatomically/physiologically) between ID and ASD, which in and of itself remains an active area of research of great relevance to many different disorders under study.

⇒ Thanks for your comments. We have revised the Conclusion and Outlook section.

Furthermore, it might be fascinating philosophically to include a brief discussion on why the polygenic developmental disorders remain at such high numbers, especially among certain populations (e.g. higher rates of autism in tech valleys). How does it make sense that some persist, counterintuitively from an evolutionary perspective? This would add interesting big picture context to your good review.

⇒ Thanks for your comments. We have added the context to our manuscript. 

No English editing required; just typos throughout. Please use a spellcheck.

⇒ Thanks for your comments. We have used a spellcheck.

Round 2

Reviewer 2 Report

Comments and Suggestions for Authors

The authors have addressed my comments and improved the overall quality of the manuscript. Thus I endorse the publication of this paper.

Comments on the Quality of English Language

Minor editing of English language required.